# Impact of Collembola on the Winter Wheat Growth in Soil Infected by Soil-Borne Pathogenic Fungi

Iwona Gruss , Jacek Twardowski * , Krzysztof Matkowski  and Marta Jurga

Department of Plant Protection, Wroclaw University of Environmental and Life Sciences, Grunwaldzki Sq. 24a, 50-363 Wroclaw, Poland; iwona.gruss@upwr.edu.pl (I.G.); krzysztof.matkowski@upwr.edu.pl (K.M.); marta.jurga@upwr.edu.pl (M.J.)

* Correspondence: jacek.twardowski@upwr.edu.pl; Tel.: +48-71-3201760

**Abstract:** The activity of some soil organisms can significantly influence the growth of plants. One of the more common are Collembola, which play an important role in suppressing soil-borne pathogens such as *Fusarium* spp. Here, *Folsomia candida* was taken for laboratory studies. The aim of the study was to assess whether springtails influence the growth of wheat and pea plants. The purpose was also to evaluate whether Collembola will reduce the occurrence of fungal diseases, presumably by feeding on fungi. The factors tested were (1) wheat grown individually or in the mixture with pea; (2) number of Collembola; and (3) the pathogenic presence of the plant fungus *Fusarium culmorum*. The experiment was carried out in four replicates for each treatment in two series. The soil used for the test was a mixture of field soil, sand, and peat. The following analyses were performed: measuring plant growth and decomposition rate, assessment of plant infection, and assessment of *F. culmorum* in springtails bodies. There was no effect of *F. culmorum* infection on plant growth, although the pathogen was present in the root neck of the plants incubated with this fungus. Collembola decreased the number of fungus colonies isolated from plants by about 45% in comparison to pots incubated without these organisms. The decomposition of plant biomass was accelerated by springtails by about 7% in the pots with moderate Collembola number. However, this was not related to improved plant growth. Additionally, *F. culmorum* was isolated from the bodies of Collembola, indicating its ability to feed on this fungus. To conclude, it was found that Collembola can decrease pathogenic fungal growth. This issue needs further studies in relation to other plants and fungus species, as well to study observed effects in the field conditions.

**Keywords:** springtails; *Folsomia candida*; wheat; pea; *Fusarium culmorum*; interactions

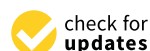

## 1. Introduction

Collembola, as representatives of mesofauna, are closely related to the soil environment [1]. Their activity in the soil indirectly affects plants. They are mediators of positive changes in the soil's structure and porosity and contribute to the decomposition of organic matter and, furthermore, the increase in the content of humus. The soil mesofauna, by stimulating humification and mineralization, provide plants with nutrients, that is, potassium, sulphur, phosphorus, iron, and, in particular, nitrogen. Collembola can also alter water infiltration to lower levels of the soil profile and therefore increase the ability of the roots to penetrate the soil [2]. Springtails regulate the activity of bacteria and fungus, including mycorrhizal fungi that live in symbiosis with plant roots. When feeding on fungal hyphae, the part of the organisms is not digested and extracted with droppings, causing its spread to other soil layers [3]. Collembola also contribute in the spreading of bacteria in the soil, including nitrogen-fixing bacteria [4]. The presence of Collembola affects plant growth mainly through their feeding activity and releasing nutrients in the environment [5].

Many species of springtails feed on pathogenic forms of fungi, thus reducing plant disease infestation [6]. Comprehensive explanations of this topic have been compiled by

Innocenti and Sabatini's [7] review work. Taking into account laboratory experiments, it was indicated that some Collembola species (*Onychiurus armatus*, *O. tuberculatus*, and *Folsomia candida)* feed effectively on plant pathogenic fungi [8]. Furthermore, it was found that plants previously infected with *Fusarium culmorum* grown in the presence of springtails (*F. candida*) had a significantly lower disease rate, as well as higher biomass than plants grown in the absence of springtails [9]. Some studies confirm that springtails feed on fungi in the environment. For example, Berg et al. [10] indicated the enzymes responsible for digesting fungal hyphae in springtails bodies sampled from the field conditions. This was confirmed using the isotope method on the Collembola forest species [11].

Fungal diseases are one of the major problems in agriculture. Currently, there are no effective biological methods that could limit them. Many groups of invertebrates, particularly Collembola, feed on fungi [12], but there is currently no practical use in reducing fungal diseases. The important pathogens, dangerous to many crops, are soil-borne fungi such as *Phytophthora*, *Rhizoctonia*, or *Fusarium*, which produce spores, sclerotia, and mycelium, constituting primary inoculum in the surface soil layer. When considering these pathogens, the more abundant the pathogen population, the greater the risk of plant infection. Therefore, in such diseases, fungus-feeding invertebrates such as Collembola could potentially decrease fungus populations [7]. The study of springtail food preferences has shown that they feed on hyphae and spores of pathogenic fungi. At the same time, it is known that the food preferences of most species are quite broad and that they can eat different groups of fungi (e.g., mycorrhizal), as well as plant tissue or bacteria. In this pot experiment, our intention was to investigate the interactions between plants, fungi, and Collembola.

This study aimed to access the effect of Collembola on the growth of plants infected by the pathogenic fungus *Fusarium culmorum*. It was hypothesized that (a) springtails will improve plant growth via the acceleration of the decomposition rate of plant biomass, (b) wheat growth will increase in the mixture with legumes more than cultivated in monoculture, and (c) springtails will reduce the occurrence of fungal diseases, presumably by feeding on fungus.

## 2. Materials and Methods

*2.1. Experimental Design*

The experiment was carried out in two series in June and September 2020. The explanatory variables were:

1. Plant: wheat, wheat, and pea;
2. Collembola number per test vessel: 100 individuals, 50 individuals, 0 individuals;
3. The presence of the plant pathogenic fungus: present, absent.

To test the effect of the factors, nine treatments were determined (Figure 1). The experiment was carried out in four replicates for each treatment (40 test vessels) in two series. The test vessels were 29 cm high, 9.5 cm × 9.5 cm wide, 11.5 cm in diameter, and had a capacity of 2600 cm$^3$, each filled with 600 g of soil. There were drainage holes in the bottom of the container, and the bottom of the container (approximately 5 cm) was covered with aluminum foil to ensure the darkening of the roots. The number of plants in the repeat was 6.

| Plant | | Wheat | | | Wheat and Pea | | |
|---|---|---|---|---|---|---|---|
| Collembola | | 0 | 50 | 100 | 0 | 50 | 100 |
| Fungus | | Present | Absent | | | | |

**Figure 1.** Description of the treatments used in the experiment.

### 2.2. Soil Preparation

The soil used for the test was a mixture of field soil, sand, and peat in a ratio of 1:1:0.5. The physical and chemical parameters of the soil are shown in Table 1. The prepared mixture was characterized by a medium rich in organic matter with a high content of calcium and magnesium and slightly alkaline. The mixture was sieved and sterilized by autoclaving.

**Table 1.** Properties of the soil used in the experiment.

| Parameter | Unit | Agricultural Soil | A Mixture Of Soil, Sand, and Peat |
|---|---|---|---|
| pH w $H_2O$ | | 5.1 | 6.6 |
| Salinity | g $NaCl/dm^3$ | <0.2 (n) | <0.2 (n) |
| $N-NO_3$ | | <10 (5) | <10 (4) |
| P | | 17 | 19 |
| K | | 30 | 14 |
| Ca | $mg/dm^3$ | 109 | 429 |
| Na | | <6 (4) | <6 (5) |
| Mg | | 17 | 47 |
| Chlorides | | 10 | 10 |
| C- org | % dry mass | 0.62 | 0.69 |
| Organic matter | | 1.07 | 1.19 |

### 2.3. Organisms and Incubation

The test fungus species was *Fusarium culmorum* (Wm.G. Sm.) Sacc. PCR analysis with species-specific primers confirmed that the tested isolate belongs to the *F. culmorum* (Supplementary Material S1). The analysis was performed in duplicate (P1 and P2). The positive control (PM) was the isolate Fc24 from the collection of UTP in Bydgoszcz, Poland.

Monoconidial cultures of *F. culmorum* were placed on potato dextrose agar (PDA) and incubated at 25 °C and a photoperiod of 12 h. The inoculum was prepared by mixing 3 plates with *F. culmorum* colonies on a PCA medium with 100 mL of physiological saline. The density of the suspension of the propagules was counted using a hemocytometer, and the desirable densities were obtained by diluting the stock with sterile water. Next, 100 mL of final suspension containing $10^5$ conidia/mL was added to the soil surface. To the remaining treatments, 100 mL of water was added. This volume of liquid was needed to obtain 50% WHC of the test soil.

The winter wheat variety (*Triticum aestivum* L.) was Natula, while the pea variety (*Pisum sativum* L.) was Cysterski. We selected these plants due to the high proportion of winter wheat in the crop rotation in Poland, while legumes are very useful as crop rotation component. In addition, both plants are easy to cultivate under laboratory conditions.

The seeds were sterilized in 10% sodium hypochlorite and rinsed with distilled water. Seed germination and early plant development were performed in Petri dishes on moist

cotton. In total, 12 seeds were placed in each Petri dish (7 cm diagonal) in combinations corresponding to the experimental design. After 15 days, the winter wheat plants had the development stage (BBCH) 12 (leaf development) and pea 2 (shoot development). Then, the additional plants were eliminated (there should be 6 in one replication). The cotton plants were placed on the surface of the soil in the test vessels.

The Collembola species used for the test was the soil-dwelling parthenogenetic *Folsomia candida* Willem, which is recommended as a model organism. Collembola were derived from the culture of the Wroclaw University of Environmental and Life Sciences. They were reared on plaster of Paris and charcoal. For the test, 12-day organisms were used. The number of Collembola per one test vessel (100 or 50) corresponds to the number of 10,000, or 5000 individuals per $m^2$, which means it is possible to see Collembola in agricultural fields. *F. candida* is able to reproduce after 2–4 weeks after hatching [13]. Therefore, at the end of the test, there should be juveniles and adults in the test vessels.

The first step of incubation was the watering of the soil in the test vessels with inoculum or water. After 3 days, the plants and Collembola were added to the test vessels in combinations and numbers according to the experimental scheme. The incubation period was 28 days. The test vessels were placed in a randomized block design in a phytotron room at 25 °C ± 2 °C under a 12 day/night photoperiod. The plants in the test vessels were watered every two days according to the water loss expressed by weight. Both series of experiments were carried out under the same conditions, including soil, inoculum volume, plant development phases, and springtails' age, as well as light and temperature conditions. Collembola were not fed during the experiment.

### 2.4. Analyzes after Incubation

*Plants.* At the end of the experiment, the plants were harvested, and the above- and below-ground parts were measured. The stage of wheat development was in the BBCH phase 28–29 (end of tilling), and the stage of development of the pea was 6 (flowering).

*Decomposition rate.* The activity of the organisms was determined by measuring the rate of decomposition. For this, the litter bags method [14] was used. Organza bags of 3 × 5 cm in diameter were used, allowing access to soil organisms. The bags were filled with 0.9 g of dry plant biomass, which was the mixture of parent grasses (*Festuca rubra* L., *Lolium perenne* L., *Poa pratensis* L., *Festuca ovina* L., *Festuca arundinacea* Schreb., *Agrostis capillaris* L.). A bag was placed in the surface soil layer in each test vessel at the beginning of the experiment and incubated for 28 days. After the incubation period, the plant biomass was separated from soil residues, dried, and weighed. The rate of decomposition was determined according to the formula [15]: $Ln(X_0/X_t) = k_t$, where $X_0$—initial biomass; $X_t$—weight loss in time t; and t—incubation time. The decomposition rate was measured only in the second series of the experiment.

*Evaluation of plant infection.* The evaluation of the plant infection was determined immediately after the removal of the plants from the soil. For this, the plant root necks were used, which are the most target plant parts. Fragments of the root necks were washed out in distilled water and inoculated on a PCA medium. Six plates were made in each combination and eight plant fragments were placed on one plate. After one week of incubation, the number of *Fusarium culmorum* colonies in each plate was evaluated. The species was identified under the microscope based on morphological characteristics [16] (manuscript Supplementary Material).

*Assessment of Fusarium culmorum in springtails bodies.* After the incubation period, Collembola were extracted from the soil in certain soil vessels using the Tullgren funnels. The extraction period was 48 h, and the Collembola was kept in distilled water. The springtails were sterilized in 100% ethyl alcohol for 10 min and transferred to PDA plates in a set of 6 plates of each combination. On each plate, springtails were placed in 6 groups of several individuals. Photographic documentation is attached in the manuscript Supplementary Material. Before placing on the plates, the group of springtails were mac-

erated using the needle. Knowing that springtails skin was sterilized prior to incubation, mainly the fungus from the intern of the body were evaluated.

After one week of incubation, the number and size of *F. culmorum* and other fungi were evaluated in each plate. Photographic documentation is attached in Supplementary Material S3. There was also a series of plates with springtails, which were sterilized to isolate the fungi on the skin of the springtails. However, the results were difficult to analyze because of contamination with other microorganisms. We established that Collembola can transfer fungi hyphae to their skin, as observed under a microscope.

### 2.5. Data Analysis

The results obtained were compared using the mixed model (proc mixed) at the significance level of $p = 0.05$. In the case of two series of experiment, the series was a repeated factor. Additionally, the HSD Tukey test was performed to indicate significant differences. The analyses were performed in Microsoft Excel and SAS University Edition. The link between experimental factors and the dependent variables was determined using constructed analysis with linear method (RDA) in Canoco software version 5.0 (Ithaca, NY, USA). The significance of the axes was analyzed using the permutation test.

## 3. Results

### 3.1. Effect on Plant Growth Parameters and Number of Fungal Colonies

*Plant growth.* We evaluated the effects of three factors that could affect plant growth: the number of plant species (1—wheat or 2—wheat and pea), the number of Collembola (0, 50, or 100 individuals), and the presence of fungus (two variants). It was found that a number of plant species were found to significantly affect both the height of the plant and the number of shoots ($p = 0.01$, $p = 0.02$, respectively). The plants were significantly higher when grown in monoculture than in coordinated cultivation with pea (Table 2; Figure 2A). Similarly, significantly more shoots developed in plants grown in wheat grown in monoculture. The other factors (Collembola and fungus) had no significant effect on plant growth. This means that inoculation with pathogenic fungi did not disturb plant development (Figure 2B).

**Table 2.** The height of the plant and the number of shoots to the number of plant species (two variants), the number of individuals from Collembola (3 variants), and the presence of *Fusarium* (2 variants). * DF—degrees of freedom; F—Mean Squared Error; $p$—significance level.

| Effect | Plant Height | | | Number of Shoots | | |
|---|---|---|---|---|---|---|
| | DF * | F | $p$ | Df | F | $p$ |
| Number of plant species | 1 | 6.12 | 0.01 | 1 | 5.47 | 0.02 |
| Number of Collembola | 2 | 0.23 | 0.80 | 2 | 0.13 | 0.88 |
| Presence of fungus | 1 | 0.01 | 0.90 | 1 | 2.48 | 0.12 |

**(A)** 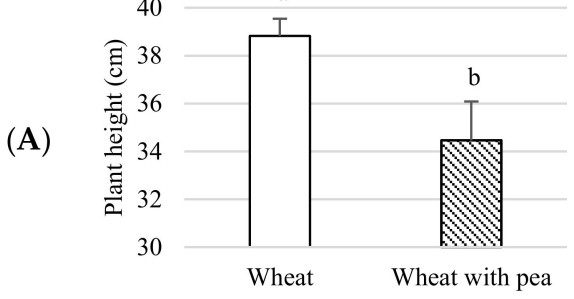

**(B)** 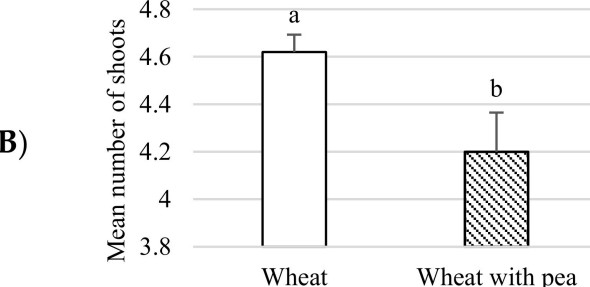

**Figure 2.** Plant height (**A**) and the mean number of shoots (**B**) of winter wheat grown individually and in the mixture with pea. The different lowercase letters indicate significant differences between treatments, $p \leq 0.05$.

*Fungus infection.* The number of *Fusarium* colonies was significantly affected both by the presence of *Fusarium* and by the number of Collembola ($p = 0.0001$, $p = 0.02$) (Table 3). The number of colonies was significantly higher in the test vessels, where the fungus was inoculated, which confirmed effective fungus infection in the tissues of plants (Figure 3A). It was also found that fungi could accidentally occur in test vessels without fungus inoculation. Taking into account the development of the effect of Collembola on the fungus, it was found that the mean number of colonies was significantly lower in the pots incubated with Collembola (both 50 and 100 individuals) compared to the test vessels without Collembola (Figure 3B). This result could indicate that Collembola could effectively decrease the development of the fungus in the soil and therefore inhibit the plant infection.

**Table 3.** Several *Fusarium* colonies to the number of Collembola (3 variants) and the presence of *Fusarium* (2 variants). * DF—degrees of freedom; F—Mean Squared Error; *p*—significance level.

| Effect | DF * | F | *p* |
|---|---|---|---|
| Number of Collembola | 2 | 4.28 | 0.02 |
| Presence of fungus | 1 | 18.49 | 0.0001 |

(A)  (B)

**Figure 3.** Several *Fusarium* colonies isolated from plants in three variants of Collembola abundance (100, 50, and 0 individuals) (**A**) and from plants previously infected with fungi and not infected (**B**). The different lowercase letters indicate significant differences between treatments, $p \leq 0.05$.

### 3.2. Collembolan Feeding on Fungus and Decomposition Rates

*Fungus colonies isolated from Collembola.* The presence of fungus in the bodies of Collembola was presented as the mean number of colonies from each treatment (Figure 4). The number of colonies was significantly higher in test vessels incubated with fungi than to control (treatments without the presence of fungus) ($p = 0.002$). This could indicate that Collembola can feed on this fungus species.

*Decomposition.* The process of decomposition of plant biomass was significantly affected by the abundance of Collembola ($p = 0.007$) (Table 4; Figure 5). The decomposition rate was significantly higher (by 7%) in the moderate number of Collembola (50 individuals) compared to the test vessels without Collembola. We can assume that the decomposition was mainly the effect of Collembola (due to the absence of other organisms in sterilized soil). Therefore, in this case, the decomposition rate is also the measure of the feeding activity.

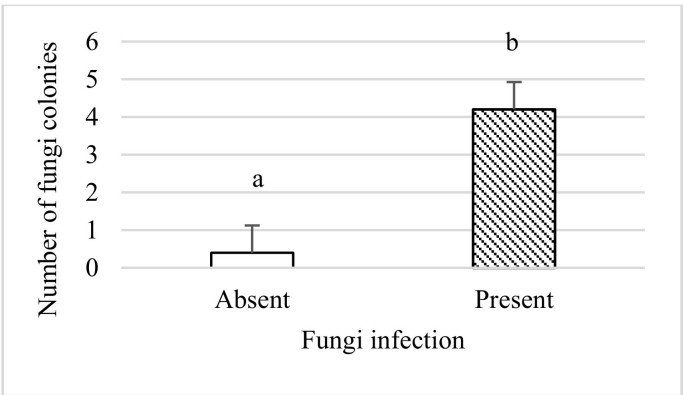

**Figure 4.** Several *Fusarium* colonies isolated from Collembola in pots incubated with (present) and without the fungus (absent). The different lowercase letters indicate significant differences between treatments, $p \leq 0.05$.

**Table 4.** Decomposition rate concerning the number of plant species (two variants), the number of Collembola individuals (three variants), and the present fungus (two variants) (mixed model). * DF—degrees of freedom; F—Mean Squared Error; *p*—significance level.

| Effect | DF * | F | *p* |
|---|---|---|---|
| Number of plant species | 2 | 0.2 | 0.82 |
| Number of Collembola | 2 | 6.44 | 0.007 |
| Presence of fungus | 1 | 0.82 | 0.37 |

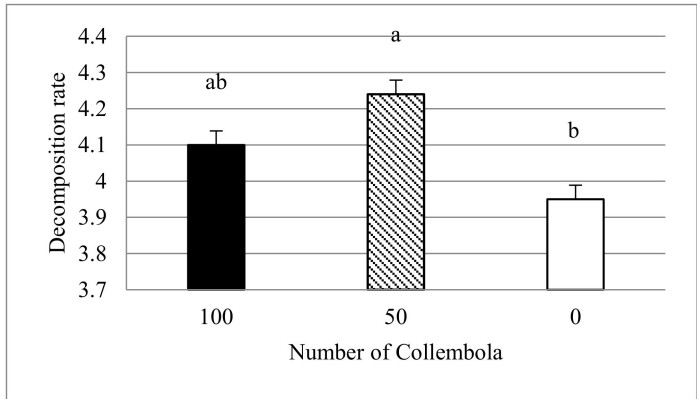

**Figure 5.** The decomposition rate in three variants of Collembola number (100, 50, and 0 individuals). The different lowercase letters indicate significant differences between treatments, $p \leq 0.05$.

### 3.3. Link between the Plant Growth Parameters, Decomposition Rate, and Number of Fungus Colonies

The trait values—dependent variables (decomposition rate, plant height, stem number and number of *Fusarium* colonies)—were correlated with the experimental factors (number of plant species, inoculated/not inoculated with *Fusarium*, number of Collembola) (Figure 6). The eigenvalues of the first two canonical axes are 0.40 and 0.11, respectively (Supplementary Material S4). The total variance explained by the two axes is 51.61% (*p* = 0.002). The most significant effects on the plot are: 1. the decrease in the height of stem number of wheat grown in the mixture and 2. the higher number of *Fusarium* colonies in treatments inoculated with the fungus. The number of Collembola has only a minor impact on the dependent variables.

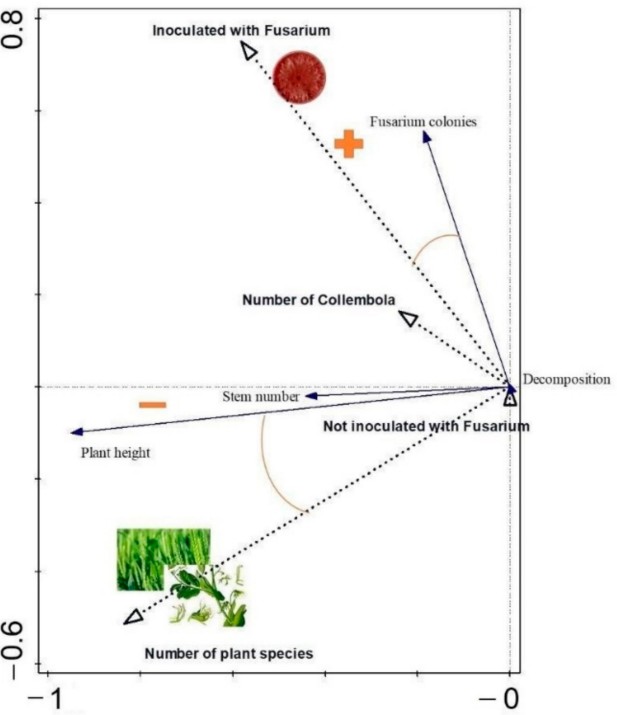

**Figure 6.** RDA biplot representing the dependent variables correlated with the experimental factors.

## 4. Discussion

The main findings of this work are:

1. Collembola decreased the number of fungus colonies isolated from plants by 45%.
2. *Fusarium culmorum* was isolated from the bodies of Collembola, indicating the ability to feed on this fungus.
3. The decomposition was accelerated by Collembola. However, this was not related to better plant growth.

We proved that Collembola decreased the number of the pathogenic fungus colonies isolated from plant tissues. Furthermore, we also demonstrated that Collembola are able to feed on fungus by isolating the fungus colonies from springtails bodies. The fungus was inoculated in the soil; therefore, Collembola probably feeds on mycelium dispersed in this environment. Thus, Collembola could decrease hyphae development and further decrease plant infection. Our result is in line with other studies conducted in a similar experiment design [6,9]. However, the study was conducted under laboratory conditions, and this ability of Collembola to feed on pathogenic fungus does not have to exist in an environment where Collembola have unlimited food choices. To date, such studies have not been performed.

At the end of the incubation experiment, the presence of *Fusarium culmorum* was confirmed in the root necks. Therefore, the infection with the fungus was effective. However, no negative effects on plant growth were observed during incubation. In this study, we placed the plants in experimental containers in the leaf development stage, while the first symptoms of *F. culmorum* infection are mostly visible before the emergence of the plant [17].

Collembola are found to regulate the decomposition processes and the nutrient cycling [3]. The rate of decomposition can be influenced by many factors, such as the quantity and quality of biomass or the size of the mesh in the litter bags [1,18]. In this study, the decomposition rate was accelerated by Collembola, but this was not related to increased plant growth. The rate of decomposition of plant biomass was higher in containers with a moderate number (50) of springtails per container compared to containers without Collembola. Our result is in line with the study by Innocenti et al. [19], where Collembola *Protaphorura armata* increased wheat growth. In other mesocosm studies [20], the presence

of springtails was shown to have a positive effect on the decomposition rate, especially in the case of increasing their species diversity. Perhaps the impact of Collembola on plant growth would be more pronounced with a longer research period. Collembola have been found to decrease the root biomass of *Poa annua* [21], which could be an explanation for the higher rate of decomposition in the moderate abundance of Collembola compared to its high abundance.

When analyzing the plant growth, it was found that the height of wheat was reduced when grown in the mixture with pea. This may be due to the strong competition between the peas and the other plant. Pea has a highly developed root system, which dominates the root system of wheat, which is the least developed of all cereals [22]. Peas could effectively absorb nutrients; an additional advantage is the ability of legumes to fix atmospheric nitrogen [23]. Endlweber and Scheu [24], who examined the influence of springtails on grasses and legumes. His experience has also shown that Collembola increases the competitive power of legumes against grasses. Furthermore, Endlweber's research [25] showed interspecific competition in the case of breeding two species of weeds in one treatment, which manifested as weaker growth.

To conclude, it was found that Collembola can decrease the pathogenic fungal development in the wheat tissues. Our study provides the basis for research on the use of springtails in biological control against fungal diseases.

**Supplementary Materials:** The following supporting information can be downloaded at: https://www.mdpi.com/article/10.3390/agronomy12071599/s1, Supplementary Material S1: The results of PCR analysis; Supplementary Material S2: Fungus colonies isolated from plants in series 2; Supplementary Material S3: Fungus colonies isolated from Collembola in series 2; Supplementary Material S4: Summary of RDA analysis.

**Author Contributions:** Conceptualization, I.G., J.T. and K.M.; methodology, I.G., K.M. and J.T.; investigation, I.G. and M.J.; resources, I.G.; writing—original draft preparation, I.G., K.M. and J.T.; writing—review and editing, I.G. and J.T. All authors have read and agreed to the published version of the manuscript.

**Funding:** The APC/BPC is financed/co-financed by Wroclaw University of Environmental and Life Sciences.

**Institutional Review Board Statement:** Not applicable.

**Informed Consent Statement:** Not applicable.

**Data Availability Statement:** Not applicable.

**Conflicts of Interest:** The authors declare no conflict of interest.

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
