# Peer review of "Impact of Collembola on the Winter Wheat Growth in Soil Infected by Soil-Borne Pathogenic Fungi"

_agronomy, doi:10.3390/agronomy12071599_

Round 1

Reviewer 1 Report

The manuscript is well written and the novelty and the purpose are clearly enounced. Material and methods is quite full.

I suggest only some re-writing and some addition to understand better several sentences and figures in the attached pdf.

A diagram of the experimental set-up would also be helpful. 

Author Response

Reviewer 1.

Thank you for the kind comments.

Comment 1. I propose, that this paragraph[h should be divided into two parts: laboratory and field.

Response: We agree, the paragraph would be easier to understand. Lines 41-51

Comment 2. Unclear the aim of the study.

Response: We improved the aim and hypotheses. Lines 67-72

Comment 3. Experimental scheme would be better.

Response: We agree and prepared an experimental diagram which is now Figure 1.

Comment 4. Figures

Response: We corrected the figures and figure captions’.

Comment 5. A sentence of perspectives.

Response: We added a sentence of perspective.

Reviewer 2 Report

The manuscript focuses on studying the effects of soil-dwelling springtails on plant growth, including the possible preventive potential of springtails against plant-pathogenic Fusarium spp. Although the subject is interesting, the manuscript, in its current status, is not suitable for publication due to significant concerns about the experimental design, the lack of quality data and adequate interpretation of the data. 

Author Response

The manuscript focuses on studying the effects of soil-dwelling springtails on plant growth, including the possible preventive potential of springtails against plant-pathogenic Fusarium spp. Although the subject is interesting, the manuscript, in its current status, is not suitable for publication due to significant concerns about the experimental design, the lack of quality data and adequate interpretation of the data.

Response: Thank you for that constructive comment. We made major changes in the results and discussion. Experimental design was updated. In our opinion the value of the manuscript strongly increased.

Reviewer 3 Report

This study involved fungi, soil collembolan and plant species, focused on the function of soil biota on plant growth and the function of soil collembolan on the fungi. The authors gained some clear results. There are some questions and suggestions, which would be helpful for the revision.

  1. It would be better for revision if you can added line numbers for the manuscript
  2. For the title showing a very big topics, thus it could be constrained.
  3. Adding a conclusive sentence at the end of abstract will be better;
  4. Why did you provide the third hypothesis here, there is no any relative contend in the Introduction.
  5. For M&M why were these two plant species selected?
  6. You isolated fungus from collembolan and conclued that the collembolan feeded on the fungi. Did you distinguish the fungi from the surface of collembolan or inside of body or the gut? Some procedures should be added in M&M.
  7. In this manuscript, some results should be discussed for the reasons. Such as the pathogenic fungi had no effect on the growth of plant, and high number of collembolan reduced the decomposition, why?

Author Response

Comment 1. It would be better for revision if you can added line numbers for the manuscript.

Response: Agree. We added the line numbers.

Comment 2. For the title showing a very big topics, thus it could be constrained.

Response: We agree. Here is the new title:

“Impact of Collembola on the winter wheat growth in soil infected by soil-born pathogenic fungi”

Comment 3. Adding a conclusive sentence at the end of abstract will be better.

Response: We added the conclusive sentence in the abstract.

Comment 4. Why did you provide the third hypothesis here, there is no any relative contend in the Introduction.

Response: Agree. We have rewritten the hypotheses which worked the abandonment the third hypothesis.

Comment 5. For M&M why were these two plant species selected?

Response: We added an explanation

Comment 6. You isolated fungus from collembolan and conclued that the collembolan feeded on the fungi. Did you distinguish the fungi from the surface of collembolan or inside of body or the gut? Some procedures should be added in M&M.

Response: We added the information to materials and methods. We distinguish the fungus inside the body.

Comment 7. In this manuscript, some results should be discussed for the reasons. Such as the pathogenic fungi had no effect on the growth of plant, and high number of collembolan reduced the decomposition, why?

Response: We have rewritten the discussion and added some comments to the results.

Round 2

Reviewer 2 Report

Regarding the revised manuscript, although some issues have been addressed and solved to some extent, the manuscript still needs a much better controlled thinking and proper terminology almost everywhere.

Author Response

Dear Reviewer,

Thank you for the constructive revision. We tried to improve our paper taking into account your comments. However, there are some issues, which we are not able to change right now such as experimental design. Please find a point-by-point description of how we have addressed the reviewer’s comments. In manuscript’s body corrections were made with track changes.

Best regards,

Jacek Twardowski

Regarding the revised manuscript, although some issues have been addressed and solved to some extent, the manuscript still needs a much better controlled thinking and proper terminology almost everywhere

Detailed Comments to the Authors

Regarding Abstract / Introduction

L20 "Collembola decreased the number of isolated from plants."

L269 "Collembola decreased the number of isolated from plants."

What did Collembola decrease?

Response: Thank you for the comment. Collembola decreased the number of fungus colonies. Fixed in the text.

L22-23 "F. culmorum was isolated from the bodies of Collembola, indicating its ability to feed on this fungus." Feeding is only a possibility! How can one distinguish between internalized and body surface absorbed fungal spores and mycelia?

Response: The Collembola were sterilized before the transfer on the PDA plates. Thus, we can suppose, that we isolated the fungus only from the intern of the body.

L24 " ... it was found that Collembola can decrease the pathogenic fungal development." What is the meaning of "pathogenic fungal development"? At what stage can Collembola interfere with the "pathogenic fungal development"?

Response: We agree, we are not able to prove the effect on fungal development with this methodology. We replaced the word “development” with “growth”. In our opinion the decrease of fungal colonies means the reduction of fungal growth.

The manuscript does not offer quality data to evaluate how the pathogenicity develops between the fungus (Fusarium) and plants. Indeed, plants seem to be already infected when springtails become available in the soil.

Response: The infection was confirmed through the isolation of fungus colonies from the root necks.

- another statement from the results section seems to contradict (L198-200)

"The other factors (Collembola and fungus) had no significant effect on plant growth. This means that inoculation with pathogenic fungi did not disturb plant development." So, based on the experiments, how can it be concluded that "Collembola can decrease the pathogenic fungal development"?

Response: In this experiment we evaluated 1. The plant growth; 2. The number of fungus colonies isolated from root necks;  3. The number of fungus colonies isolated from Collembola bodies.

In our opinion the Statement "Collembola can decrease the pathogenic fungal development/growth" is truth, because as presented on Figure 3 A, Collembola decreased the number of colonies in the root necks of infarcted plants, thus suppressed the growth of fungus.

L35-36 - mycorrhizal fungi are not "microorganisms", they are visible

Response: We agree. The current text is: “Springtails regulate the activity of bacteria and fungus, including mycorrhizal fungi that live in symbiosis with plant roots”.

 L38-41 - this section has several confusing details

- "Feeding mycorrhizal fungi can cause (?) the growth of soil bacteria" - How? The interaction between fungi and bacteria must be specified.

- "soil bacteria bind to (?) nitrogen ... "

- " ... springtails .... altering the microorganisms"? How and in what sense ? What does "altering" mean?

Response: We are sorry for those confusions. We corrected the text as followed: “Springtails regulate the activity of bacteria and fungus, including mycorrhizal fungi that live in symbiosis with plant roots. When feeding on fungal hyphae, the part of the organisms is not digested and extracted with droppings, causing it’s spread to other soil layers [3]. Collembola also contribute in spreading of bacteria in the soil,  including nitrogen-fixing bacteria [4]. The presence of Collembola affects the plant growth mainly through their feeding activity and  release nutrients in the environment [5].”

Tables

- several abbreviations (F, DF, Df) are not explained in the text

Response: We added the abbreviations’ explanations to the tables.

Materials and Methods / Experimental Design

The experimental setup is so simple but the way the experiments are described and presented is unacceptable.

- " ... factors were tested: 1. Plant diversity" What kind of "plant diversity" was tested?

Response: We replaced the sentence “…factors were tested” with “The explanatory variables were”. We agree – “plant diversity” is not correct in this case. We use the word “plant” instead.

- Figure 1 - by just looking at the details of the figure itself it is impossible to find out how Collembola and fungus were combined

Response: In our opinion it is possible. For wheat we had differentiated Collembola number (0, 50, 100) and pots, with fungus inoculation and no inoculation.

L164-172

- "springtails were sterilized in 100% ethyl alcohol ... " - how long were they incubated ?

Response: The duration of the sterilization was 10 minutes. Added to the text.

L168 - unclear sentence

Response: The sentence corrected.

L170--172 It is still unclear how the surface adsorbed and internalized fungal remains / cells could be differentiated

- "knowing that springtails skin was sterilized" - it is not an experimentally supported evidence

Response: In our opinion, we isolated mainly the fungus from the intern of the body. We added the word “mainly”.

Results

- The plant growth data is not a real result!

Response: The stage of wheat development was in the BBCH faze 28-29 (end of tilling) and the stage of development of the pea was 6 (flowering). Collembola and fungus had no significant effect on plant growth. This means that inoculation with pathogenic fungi did not disturb plant development

L212-213 How the higher number of fungal colonies recovered from the vessel itself could confirm the infection of the plants? This is not a direct evidence.

Response: Fungi were isolated from tissue, not from vessels. Therefore, the number of colonies present in plants indicates the level of infection.

Figure 3 - The number of recovered colonies are really low and I am very sceptical about the real significance

Response: The method assumes a simple criterion of infection assessment, the number obtained in the colony tissues. These results were obtained and are statistically significant and conclusiveness of the data.

L242-243 "we can assume that the decomposition was mainly the effect of Collembola"

- I wonder if the introduced Collembola was perfectly free of microbial partners

Response: We assume, that Collembola were mostly responsible for the decomposition. The soil and seeds were sterilized before the experiment. The Collembola were cultured in the climatic chamber, with any other organisms. We cared on the condition of the colonies.

- The soil should have been tested for possible "Collembola dependent" incoming microbiota, by just running amplicon sequencing on various body parts of the springtails and then testing the soil samples

Response: Thank you for this advice. For now the idea it is not possible to carry out. We will use the sequencing on various body parts of the springtails in the future research.
